



# Physical Experiments on the Development of an Ice Tunnel from an Upstream Water Reservoir through Simulated Glacier Dam

Chengbin Zou[1], Paul A. Carling[1,2], Zetao Feng[1], Daniel R. Parsons[3], Xuanmei Fan[1*]

[1]State Key Laboratory of Geohazard Prevention and Geoenvironment Protection, Chengdu University of Technology, Chengdu, Sichuan 610059, China
[2]Geography & Environment, University of Southampton, Southampton, SO17 1BJ, UK
[3]Energy and Environment Institute, University of Hull, Kingston-upon-Hull, HU6 7RX, UK

*Correspondence to: Xuanmei Fan (fxm_cdut@qq.com)

**Abstract.** The hydraulic and glaciological conditions that control the timing and mode of floods from proglacial-lakes impounded by glacial-ice fronts are not well known. Yet, flooding due to the failure of such lakes is increasing due to climate change, posing increased risk to people and infrastructure downstream. Many floods occur due to tunnels developing within the ice, allowing impounded water to discharge to the glacier margin rapidly. However, the manner in which ice tunnels develop through time is understood poorly; current understanding being conditioned by referral to physical theory of ice melt in the presence of flowing water and limited field observations. In this study, the basic principles of the development of a simple linear ice tunnel are explored in a laboratory flume. A sudden flux of water from an upstream water reservoir passes through an open circular tube formed within an ice block. Growth in the shape of the tube simulates the development of an ice tunnel within a glacial dam. The velocity within the entrance to the tunnel and the discharge through the tunnel were recorded, as the head within the reservoir reduced. The data are used to determine the temporal development of the energy slope, and the roughness, size and shape of the tunnel for different water temperatures and flood durations. An increase in the water temperature was a significant positive control on the rate of rise of hydrographs. The behaviour of the energy slope can be defined using the pipeflow equation proposed by Barr in 1981, with the Nikuradse equivalent roughness value increasing from $c.$ $10^{-9}$ to $10^{-4}$ m as the hydrograph progresses. For any given temperature and for skin roughness conditions, the surcharged wetted tunnel cross-sectional area increases as a logarithmic function of time whilst velocity also increased on the rising hydrograph. As frictional melt induces form roughness, velocity declines and the surcharged tunnel cross-sectional area increases to accommodate the discharge. Once a free surface occurs within the tunnel on the falling hydrograph limb, the decline in the open-channel wetted area with time is linear. The initial circular tunnel section enlarges as an ovoid as the surcharged discharge increases. Once a free surface develops, downcutting is pronounced, leading to a final key-hole shape to the tunnel. Incorporating a time-varying Manning roughness coefficient, the simplified Nye model for discharge through an ice tunnel reproduced well the observed surcharged-tunnel discharge data.

## 1 Introduction

Proglacial lakes, impounded near the snouts of valley glaciers, are increasing in number and extent within mountainous and polar regions around the globe in relation to changing climate (Carrivick *et al*., 2020; Sugar *et al*., 2020; Emmer *et al*., 2022). Such lakes provide reservoirs that may sustain river flow for water resources (Huss *et al*., 2017; Vuille *et al*., 2018) and at the same time pose a threat to communities and infrastructure downstream should the ice-dams fail catastrophically (Clague *et al*., 2012; Haeberli *et al*., 2016; Vuille *et al*., 2018; Veh *et al*.,



2020; Stuart-Smith *et al*., 2021). Catastrophic failures result in glacial-lake outburst floods (GLOFs), also known as "jökulhlaups", which can have significant downstream impacts (Carrivick and Rushmer, 2006; Carrivick and
Tweed, 2016; Cook *et al*., 2018; Dubey and Goyal, 2020; Thompson *et al*., 2020; Veh *et al*., 2020).

Ice dam failure can occur through a variety of means (Carrivick *et al*., 2017), including over-topping by rising lake level or by a wave of translation — the latter due to a landslide of ice or rock entering the lake – over-topping the dam. Overtopping may lead to thermal down-cutting of a supraglacial channel through which the lake empties either partially or fully. Alternatively, a rising lake level can exceed the ice overburden pressure inducing
mechanical fracturing and complete or partial dam collapse. More usually, lake water pressure causes the opening of water conduits through the barrier. More generally, fracturing and the development of other cavities can be induced through the development of crevasses, by glacier motion and deformation unrelated to lake level changes (Kingslake, 2013). Fracture and cavity networks can form either complex series of partly-connected conduits (Guillet *et al*., 2022), or a single conduit (Fig. 1) may result from lake to outlet (Dussailant *et al*., 2010; Bazai *et*
*al*., 2020). Once flooded, these single or conduit networks can evolve as water-eroded tunnels either englacially (Colgan *et al*., 2016), or at the base of the ice-dam where the substratum is rock or glacial diamicton and the roof of the tunnel is ice (Gulley *et al*., 2014; Gimbert *et al*., 2016). The little researched situation where sub-glacial water pressure can cause the ice over-burden to lift resulting in an irregularly-shaped broad conduit (Wortmann *et al*., 2014; Einarsson *et al*., 2017) is not considered herein. Similarly, tunnels adjacent to a rock wall (Walder &
Costa, 1996), perhaps at the lateral margin of the glacier, are not considered here.

As the temperature of the flood water is above freezing-point, the englacial fully or partially-flooded conduits are enlarged primarily by thermal erosion of the ice induced by water flow, although structural collapse may also play a role locally. In certain circumstances, although not all, hydraulic fracturing and basal ice melt at the rock bed surface may play a role in conduit development (Fowler, 1999). Over time-scales usually greater than that of
individual GLOFs, ice deformation can close conduits or reduce conduit volume and connectivity (Harper and Humphrey, 1995; Fountain and Walder, 1998; Jarosch and Gudmundsson, 2012). However, at time scales shorter than that of a GLOF, closure or an increase in tunnel capacity is more likely due to structural collapse of the ice tunnels, which may impede or enhance the efflux of a GLOF, modifying the hydrograph. Without tunnel closure, the thermodynamic result of floodwater flow through an ice tunnel should result in a rapid, near-monotonic rise
in the discharge of the GLOF through time (Nye, 1976; Spring and Hutter, 1981; Clarke, 1982) amenable to numerical calculation and simulation (Carrivick *et al*., 2020; Bazai *et al*., 2022).

Whereas complex tunnel systems might be considered in the future, the focus of the present study is the evolution of a simple straight englacial tunnel responsible for the emptying of a water reservoir (lake) under controlled experimental conditions. The purpose of the experiments is several-fold. Firstly, a straight conduit is
an end member geometry of more complex networks that needs to be understood before complexity is introduced to experiments. Further, given the simple geometry selected, the forms of the hydrograph and associated velocity curve are defined in a general sense from the empirical data and can be compared with published theory pertaining to ice-dam tunnel flows. The effect of varying the lake water temperature on the flood wave properties is assessed. The hydraulic ice roughness is estimated, as is the behaviour of the energy gradient through time. The development
of the size and shape of the tunnel is detailed through time and distance along the tunnel. In this manner, we provide experimental data that may assist in the design of field-sampling campaigns exploring natural ice-tunnels and GLOFs, as well as in the refinement of theory.



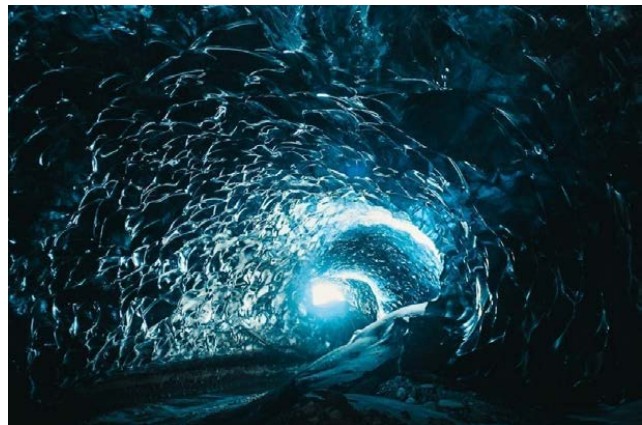

**Figure 1.** Ice tunnel beneath the Vatnajökull glacier (Iceland), formerly surcharged by summer meltwater but naturally drained during the winter **(https://visitvatnajokull.is/ice-caves-in-vatnajokull/#:~:text=Vatnaj%C3%B6kull%20Region%20is%20a%20great,glaciers%20in%20Europe%20by%20volume.** Diameter of tunnel c. 3─6 m. Note the scalloped ice walls provide form roughness when the tunnel is filled by flowing water.

## 2 Method

### 2.1 Experimental design

Experiments were undertaken in simplified conditions using a rectangular upstream water reservoir discharging through an initially straight, horizontal, circular tunnel within a rectangular block of ice (Fig. 2). The reservoir had a width of 0.6 m and a length of 2.5 m with a maximum volume of 0.9 m$^3$. The complex structure of glacier ice (Shumskiy, 1960; Boulton, 1972) is very difficult to simulate physically (Glen, 1955; Iverson, 1990) so we utilized pure water ice on the basis that the behaviour of the two media during melt would be similar, if not identical. Pure water ice has a density of 0.917 kg m$^3$ (Cuffey and Paterson, 2010), which is the maximum anticipated for pure glacier ice near the glacier surface ($\cong$ 0.900 kg m$^3$ $\cong$ 0.917 kg m$^3$). At a depth of over 1km, the density of glacier ice $\gneqq$ 0.921 kg m$^3$ but, although such ice thicknesses may have been approached in the case of Quaternary ice dams, they are not associated with modern ice dams. The ice-block ($H = 0.65$ m high; $B = 0.6$ m wide and $L = 0.5$ m long) was created by freezing clear still-water in place between the walls and above the bed of the reservoir which were refrigerated in the vicinity of the ice-dam. Temporary vertical partitions delimited the upstream and downstream extent of the dam and were removed once the water was frozen.

GLOFs manifestly are highly turbulent, reflected in high Reynolds numbers ($Re$) (Carling *et al.*, 2010). To achieve a degree of dynamic similitude, the Reynolds numbers of the tunnel flows were designed to be fully-turbulent ($10^4 < Re < 10^6$). A horizontal 9 mm diameter circular copper tube was blocked at each end, fixed to the partitions and frozen into the ice dam at an axial height of 0.25 m ($h_c$) above the centre of the flume bed. The reservoir was filled quickly to a depth ($h_o$) of 0.6 m with water from the city water supply and the partitions then removed. The tube was extended by a threaded joint at the downstream end and a gentle hammer blow freed the tube from the ice and the tube was withdrawn. Discharge commenced when the tunnel (diameter: $d = 10$ mm) was opened by the final tube withdrawal.



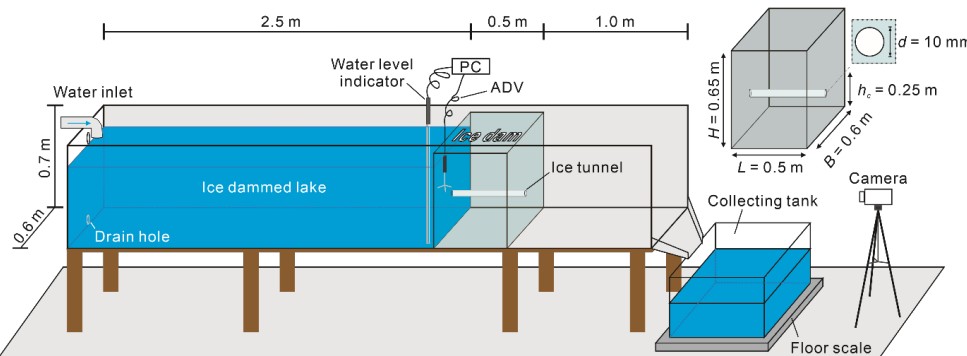

**Figure 2. Definition diagram of the experimental facility**

The consequent water level ($h$) drop in the reservoir was recorded at 1mm resolution at a frequency of 20 Hz using an YWH200-D digital wave probe (Chengdu Xinda Shengtong Technology Co., Ltd). The discharge for a

given time increment ($Q_t$) through the tunnel was determined using two complementary methods. Firstly, the influent discharge ($Q_t$IN) was calculated through the change in storage ($S$) (Eq. (1)) over time ($t$). Secondly, the effluent discharge ($Q_t$OUT), exiting the flume through the ice tunnel, was collected in a sump tank placed on a Sichuan Langke Seiko Echnology Co. Ltd recording weighing scale. The effluent discharge was weighed cumulatively and continuously at 2 s intervals with a resolution of 0.25 g. As expected, the two methods gave

consistently comparable results (Fig. 3). The outflow was marginally greater than the inflow (< 1 %), reflecting the additional small quantity of water eroded from ice-melt at the upstream ice-dam margin (*c.* 20 mm recession during the experiment with the warmest water) and the enlargement of the ice-tunnel (detailed in the Results). In the Results, the effluent discharges are utilized, with data archived as a Supplement.

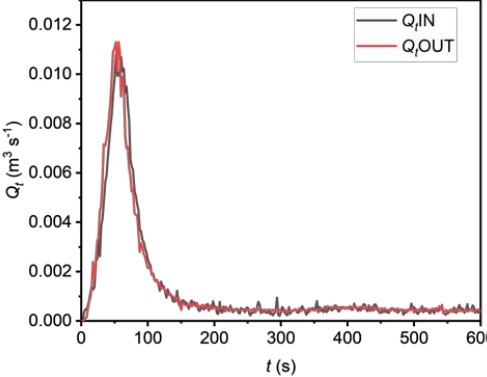

**Figure 3. Comparison of inflow to ice tunnel and the outflow from the ice tunnel.**

The distribution of the cross-sectional average velocity ($\bar{U}$) at variable distances ($x_i$) from the tunnel inlet as time varies throughout the tunnel wetted cross-section ($A$) is unknown. We estimated a velocity using a Nortek Vectrino™ Acoustic Doppler velocimeter (ADV), recording at 20 Hz, on a rigid mount close to the tunnel entrance (but not obstructing the flow) with the focus of the beams just inside the tunnel entrance at a distance $x_o = 0 + \varepsilon$,

where $x$ is the distance along the tunnel (from the vertical upstream ice-wall) and $\varepsilon$ indicates a small unmeasured additional length. Due to turbulence, the velocities varied rapidly through time and were smoothed and resampled



for plotting and further analysis. Assuming the velocity at $x_o = 0 + \varepsilon$ is indicative of the average velocity ($\overline{U}_{x_o}$) in the entrance to the surcharged tunnel, the wetted entrance-section cross-sectional area ($A_{ot}$) as time increments is given by Eq. (5). Video records were used to check for tunnel growth at the entrance and exit and for evidence of

the tunnel being surcharged or not. Preliminary visual observations showed slight funnel-shaped widening of the entrance such that the tunnel further inside the ice dam likely was of smaller cross-sectional area. This latter observation is pertinent to the interpretation of the Results and is considered further in the Discussion.

Water temperature of ice-dammed lakes often is assumed to be close to zero although temperature of Icelandic GLOFs have been measured, for example, as 0.05 °C (Rist, 1955), 0 to 1°C (Einarsson *et al.*, 2017) and

up to 10 °C (Carrivick *et al.*, 2020). Thus, the effects of water temperature were explored by running experiments at two different time of the year as the water temperature of the city water supply varied by season. We included a higher water temperature as well as a low value to identify clearly the effects of temperature on tunnel growth. For the higher temperature, three nominally self-similar experiments were conducted using identical initial conditions for a steadily falling reservoir. This *unperturbed* condition was varied by introducing a pulse of

additional flow to a fourth experiment yielding a *perturbed* condition. In the latter case, an additional pulse of water was added to the hydrograph to simulate such a variation in a natural discharge. These higher temperature experiments were run at temperatures of 21.5 °C, 23.5 °C, 24.0 °C and 25.2 °C in late summer of 2021 and lower temperature experiments at 11.2 °C and 12.6 °C were conducted in the winter of 2021, with a 15.2 °C run in the spring of 2022. Although controlled temperatures would have been preferred, we did not have the means to cool

the reservoir in a controlled manner. So, water temperatures were not controlled but represented the ambient temperature of the city water supply.

**2.2 Calculation of discharge function**

The ice dammed lake is simulated as a tank with vertical walls. The discharge ($Q$) on the falling limb of such a reservoir through time ($t$) is directly proportional to the storage ($S$) represented by the change in the reservoir

water level ($h$), mediated by the recession constant ($k$) as described by the Eq. (1). Considering mass conservation, the linear reservoir is expressed by the differential Eq. (2), resulting after solution in the exponential Eq. (3), with $k$ defined by Eq. (4) and the subscript 1 indicates a time step after the start time $t_o$:

$$Q = kS \, , \tag{1}$$

$$Q = \frac{dS}{dt} \, , \tag{2}$$

$$Q_1 = Q_o \, e^{-k \, (t_1 - t_o)} \, , \tag{3}$$

$$k = \frac{\ln Q_o - \ln Q_1}{t_1 - t_o} \, . \tag{4}$$

A note is required at this point based on initial trials, so that both the Method and the Results are understandable. Complexity is introduced to a simple mass balance by the fact that the tunnel is subject to thermal erosion, which can be rapid due to the high water temperature (*e.g.* 21.5 °C) and the cross-sectional area of the tunnel increases

as time increments. Initially, the tunnel is filled by the discharge as surcharged pipeflow. Not only does the cross-section of the tunnel increase in area, but the base of the tunnel erodes downwards, until (in principle) the floor of the flume is reached to give a flat base to the tunnel. At some time, the water level in the reservoir falls to the top of the tunnel allowing air to penetrate the full length of the tunnel above open-channel flow. At this time, the ADV





stops recording the velocity as it becomes emergent. From this point in time, although the tunnel initially remains

fairly full, the cross-sectional area of the flow is less than the cross-sectional area of the tunnel although discharge
is conserved in accord with Eq. (3). Latterly the cross-sectional area of the tunnel will be greater than the wetted
area as the tunnel base incises until the tunnel is 'dry', which occurs when the water level in the reservoir is equal
to the height of the base the tunnel inlet. Application of Equations (1) to (4) include the additional discharge due
to ice-melt at the dam face and within the tunnel which together constituted < 1 % of the reservoir volume, as

noted above.

**2.3 Calculation of energy slope, roughness, tunnel cross-sectional area and Reynolds number**

Three important hydraulic parameters remain to be determined for the tunnel flow. These are the hydraulic
gradient ($I$), the hydraulic roughness of the tunnel ($k_s$) and the equivalent diameter ($d$) of the wetted tunnel section
as a function of time.

The hydraulic gradient is the difference between the height of the water surface in the reservoir and the height
of the water surface at the tunnel outlet. The former was measured, but the latter can only be estimated from visual
observations and video recordings, so $I$ is calculated instead. The equivalent diameter is obtained assuming the
evolving tunnel wetted cross-section is circular (until final draining induces down-cutting), as the average wetted
cross-sectional area ($\overline{A_{ot}}$) and average velocity ($\overline{U}$) at the entrance to the tunnel are known for time increments:

$$A_{ot} = {Q_t}/{U_t} \, , \tag{5}$$

and $d$ for any time increment is:

$$d = 2\sqrt{\frac{A_{ot}}{\pi}} \, . \tag{6}$$

The hydraulic Nikuradse equivalent (skin) roughness ($k_s{'}$) of the tunnel effectively should be constant (Colebrook
& White, 1937), as the medium does not vary. However, shape changes in the tunnel due to melt might result in

some variation in $k_s$ as form roughness ($k_s{''}$) develops (Fig. 1); thus $k_s = k_s{'} + k_s{''}$. Given the above, the variation
in $I$ was calculated iteratively for assumed values of $k_s$ utilizing the Barr (1981) approximation of the Colebrook-
White equation for fully-turbulent pipeflow:

$$\frac{0.9003}{d^2\sqrt{gdI}} = -1.9log_{10}\left[\frac{k_s}{(3.71d)^{1.053}} + \left(\frac{4.932\upsilon d}{Q}\right)^{0.937}\right]. \tag{7}$$

Iteration continued until the calculated values of $I$, as a function of time, fell within the geometric constraints on

the energy slope, as follows. Equation (7) applies to surcharged tunnels and so should apply well to the initial
draining conditions. Once the water level in the reservoir falls to the level of the invert in the entrance tunnel, air
enters the tunnel and open-channel flow occurs. At first, the free-surface will be small and Eq. (7) should still
apply reasonably well, but as the free surface increases in area Eq. (7) will apply less well. Such deviations in
performance of Eq. (7) should be evident as deviations in estimates of $I$ with time; nevertheless calculations were

terminated shortly after the time when visual and video observations indicated open-channel conditions were
developing at both the entrance and exit to the tunnel.

To ensure fully-turbulent flow pertained, as was noted above, the variation through time in the values of the
Reynolds number just within the tunnel entrance for each flood was determined as follows:



$$Re = \frac{\bar{U}_{x_0}\, d}{\nu}, \tag{8}$$

where ν is the kinematic viscosity of the water for the measured water temperatures.

### 2.4 Determining the shape of the ice tunnel

The unperturbed discharge experiment was repeated for the range of water temperatures, but the flow was stopped abruptly at 46 s, 75 s and 126 s, as explained below. The tunnel remained surcharged for the 46 s and 75 s runs but air had entered the tunnel for the final 126 s run. For the 46 s and 75 s cases, once air entered the tunnel, the

experiment was terminated abruptly by blocking the upstream end of the tunnel with a partition and by rapidly draining the reservoir via a tap. The purpose of different run times was to explore the effect of time and water temperature on tunnel development. Although we could not control the temperature, the range of experimental conditions did provide insight into the role of these two parameters. Immediately after draining, commercially-available polyurethane (insulation) foam was injected into each 'dry' tunnel; this foam expands by around 30

times its initial volume and then sets hard. The purpose was to mould the tunnel shape. Distortion occurred at the entrance and exit but the form of the main tunnel was reproduced, albeit with a presumed little unmeasured melt around the tunnel margins. The mould was sliced to provide cross-sections. The area of each cast cross-section ($A_c$) was measured.

### 2.5 Modelling the observed discharge hydrographs

The expected hydrographs were reproduced using the simplified Nye (1976) model of water flow through an ice tunnel, as presented by Carrivick *et al.* (2017). The model is appropriate as it relates to short tunnels of simple section. The novelty of our application is that both the measured variation in the energy slope, the enlargement of the tunnel due to ice melt and the variation in the roughness parameters of the observed discharge hydrograph could be used to modify the related model parameters and so adjust the modelled hydrographs. Time-dependent

viscous closure of the tunnel was neglected due to the small scale of the experimental ice block and the short times involved in the discharge events.

### 3 Results

### 3.1 Unperturbed discharge experiment (temperatures: 21.5 to 25.2 °C)

These experiments provided almost identical results so here we report only the results for a temperature of 25.2 °C.

For the first 32 s, the flow velocity increases steadily (Fig. 4) as an exponential function (equation not illustrated: $r^2 = 1$), with Reynolds numbers of $10^5$. This rapid increase in velocity is facilitated by the rapid erosion of the tunnel from a cross-sectional area of $7.854 \times 10^{-5}$ m$^2$ at time zero to a sectional area of *c.* 0.027 m$^2$ within the first few seconds. During this time, the discharge also increases steadily and roughly monotonically (Fig. 4). The discharge peaks at 52 -58 s, subsequent to which the falling hydrograph is described by Eq. (3) where $k = -0.043$

($r^2 = 1$) until 124 s.

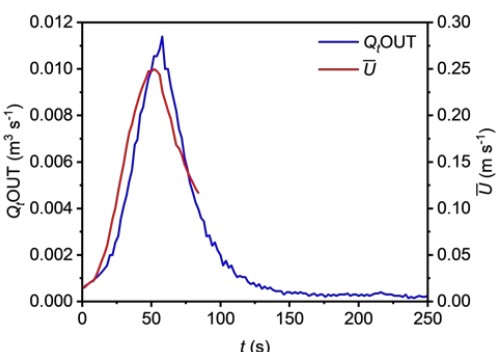

**Figure 4. Variation in discharge and velocity with time for an unperturbed hydrograph.**

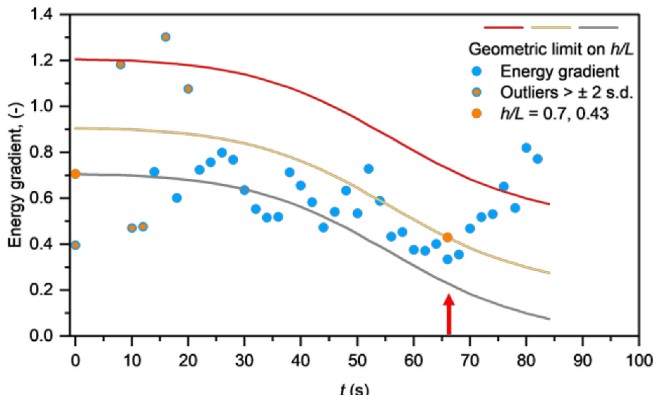

**Figure 5. Energy gradient calculated using Eq. (7), as a function of time.** The trendline for the blue dots shows a decrease
in the gradient as the reservoir empties, despite down-cutting of the tunnel into the basal ice. The grey curve represents the
geometric limit on h/L if the base of the ice tunnel is constant at 0.25 m above the flume base, which is the initial height of the
tunnel entrance. The yellow curve represents the geometric limit on $h/L$ if the base of the ice tunnel is constant at 0.15m above
the flume base, which is the approximate height of the base after down-cutting until the flow in the tunnel becomes aerated,
with a free water surface. The red curve represents the geometric limit on $h/L$, if the base of the ice tunnel is equal to the base
of the flume – which occurred at a time > 115 s. Orange dots indicate the initial $h/L = 0.7$, and when $h/L = 0.43$; i.e. just before
the tunnel aerated (red arrow) fully at $t = 66$ s.

The initial water depth ($h$) above the tunnel inlet was 0.35 m, and the ice dam length ($L$) is 0.5 m, so the
initial energy slope, $I = h/L \approx 0.7$. As water depth in the reservoir falls, there was an unmeasured reduction in base
level at the exit from the tunnel as the ice was eroded at the base of the tunnel; nevertheless, $H/L$ should be $< 0.7$
as the reservoir empties. From $t = 14$ s and for $k_s = 1 \times 10^{-9}$ m increasing to $k_s = 1 \times 10^{-6}$ m, Eq. (7) returns $\bar{I}$ values
decreasing in value over time; which range of estimated values is considered an acceptable indicator of the
dynamic energy slope when compared with an initial value of $c$. 0.7 (Fig. 5). It is evident that the energy gradient
at first is equivalent to that expected given the initial height of the tunnel above the base of the ice dam, and then
trends across a line which represents a tunnel that has eroded downwards by 0.10 m, and finally from $c$. $t = 48$ s
and for $k_s = 1 \times 10^{-4}$ m the energy gradient becomes equivalent to that expected if the base of the flume is exposed

(Fig. 4), as occurred towards the end of the experiment. The roughness of the tunnel increased during the discharge event from about $k_s = 10^{-9}$ to $10^{-4}$ m.

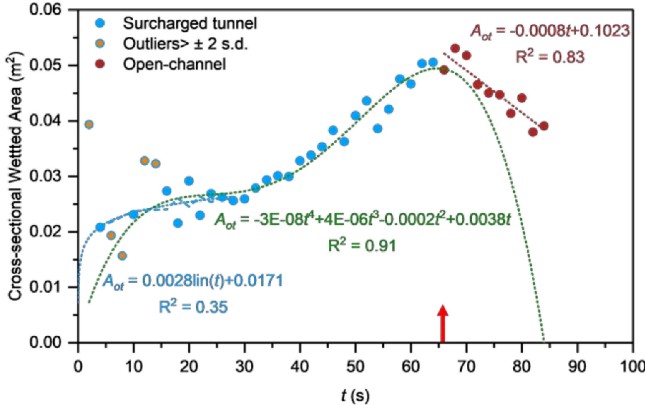

**Figure 6. Increase in the wetted cross-sectional area ($A_{ot}$) of flow with time.** Dashed blue line indicates initial data trend that is defined by an initial rapid logarithmic increase in tunnel area during initial surcharge (blue dotted curve). Data scatter precludes defining a simple continuous function to link the initial logarithmic function to a 4th-order polynomial latterly (green dotted curve). The tunnel became aerated (red arrow) fully at $t = 66$ s. Final draining of the tunnel is represented by a linear function (brown dotted curve). Fitted functions exclude outliers.

Roughly 2/3 of the frictional heat is available for melting (Roethlisberger, 1972), and so there is an expectation that the tunnel will rapidly enlarge. After opening the tunnel at $t = 0$ the cross-sectional area of the nominal 0.01 m diameter entrance tunnel increases from $7.854 \times 10^{-5}$ m$^2$ to $A_o \approx 0.023$ m$^2$ within a few seconds (Fig. 6). The cross-sectional area data initially are scattered due the turbulence-induced variation in the mean flow velocity and the unmeasured variations in the tunnel boundary roughness; an issue considered below. Nevertheless, certain characteristic behaviour is evident. For the first 28 seconds during which flow velocity was increasing, the velocity is sufficient to evacuate the discharge with minimal increase in the cross-sectional area ($\overline{A_{ot}} \approx 0.025$ m$^2$ for $30 > t > 4$ s) and the tunnel is surcharged. During this time $\overline{A_{ot}}$ is increasing in size due to melt all around the boundary; the change in cross-sectional area being best described by a logarithmic function of time. Thus, after the initial erosion, a straight circular surcharged tube characterized by skin roughness alone, suffices for conveyance. After $t = 30$ s the rate of velocity increase declines and $\bar{A}$ increases to accommodate the discharge. The increase in $\overline{A_{ot}}$ occurs as irregularities in the ice-walls develop, increasing the form roughness by down-cutting such that shear stress on the upper invert reduces. Air entrainment likely occurred before it was visibly observed, as down-cutting eventually led to a free-surface from which time $\overline{A_{ot}}$ is less than the area of the thermally-eroded ice tunnel. Final draining of the tunnel occurs after 66 s and is well-described by a simple linear function. Although the complete variation in tunnel cross-sectional area through time is complex, the general trend is commensurate with the form of a fourth-order polynomial, as illustrated by the green curve in Fig. 6.

### 3.2 Perturbed discharge experiment (temperatures: 21.5 to 25.2 °C)

For the first 34 s, the flow velocity increases steadily (Fig. 7) as an exponential function (not shown: $r^2 = 1$). This rapid increase in velocity is facilitated by the rapid erosion of the tunnel from a cross-sectional area of $7.854 \times 10^{-}$





$^5$ m$^2$ at time zero to a sectional entrance area of c. 0.027 m$^2$ within the first few seconds. Thereafter the wetted

cross-sectional area continues to increase exponentially. Apparent fluctuations in the wetted cross-sectional area are due to the unsteady velocity and the unmeasured variations in the tunnel boundary roughness, an issue considered below. During this time, the discharge also increases steadily (Fig. 7). The discharge peaks at 52 -56 s, subsequent to which the falling hydrograph is described by Eq. (3) where $k$ = -0.016 (r$^2$ = 0.96) until 64 s. The addition of two pulses of water into the reservoir at 64 s and 78 s causes a shoulder in the discharge curve and $k$

= -0.012 (r$^2$ = 0.90) between 68 s and 102 s. After 104 s, recession is steady and characterized by $k$ = -0.025 (r$^2$ = 0.96).

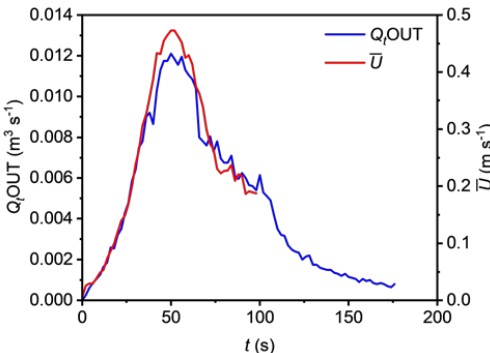

**Figure 7. Variation in discharge and velocity with time for the perturbed hydrograph**

The initial water depth ($h$) above the tunnel inlet was 0.35 m, and the ice dam length ($L$) is 0.5 m, so the

initial energy slope, $I$ = $h/L$ ≈ 0.7. As water depth in the reservoir falls, there was an unmeasured reduction in base level at the exit from the tunnel as the ice was eroded at the base of the tunnel; nevertheless, $h/L$ should be < 0.7 as the reservoir empties. From $t$ = 4 s until the 98 s, and for $k_s$ = 1 x 10$^{-10}$ m, Eq. (7) returns $\bar{I}$ = 0.56; s.d. = 0.09, decreasing in value over time; which range of estimated values is considered an acceptable indicator of the dynamic energy slope when compared with an initial value of $c$. 0.7 (Fig. 8). It is evident that the energy gradient

at first is equivalent to that expected given the initial height of the tunnel, and then trends across a line which represents a tunnel that has eroded downwards by 0.10 m, and finally the energy gradient approaches that expected if the base of the flume is exposed (Fig. 8), as occurred towards the end of the experiment. The roughness of the tunnel was fairly constant: $k_s$ = 10$^{-10}$ m.

After opening the tunnel at $t$ = 0 the cross-sectional area of the nominal 7.854 x 10$^{-5}$ m diameter tunnel

increases at the entrance to $A_{ot}$ ≈ 0.025 m$^2$ within a few seconds (Fig. 9). The cross-sectional area data are scattered due the turbulence-induced variation in the mean flow velocity. Nevertheless, certain characteristic behaviour is evident. For the first 52 seconds during which flow velocity was increasing, the velocity is sufficient to evacuate the discharge with minimal increase in the cross-sectional area; $\overline{A_{ot}}$ ≈ 0.026 m$^2$ for 52 > $t$ > 6 s) and the tunnel is surcharged. During this time $\overline{A_{ot}}$ increases slowly, the detail being lost in the noise; the channel

cross-sectional area is best described by a logarithmic function of time. Thus, after the initial erosion, a straight circular surcharged tube characterized by skin roughness alone, suffices for conveyance. After $t$ = 52 s velocity declines and $\overline{A_{ot}}$ increases to accommodate the discharge. The increase in $\overline{A_{ot}}$ occurs as irregularities in the ice-walls develop increasing the form roughness by down-cutting such that shear stress on the upper invert must reduce. Air entrainment likely occurred before it was visible observed, as down-cutting eventually led to a free-


surface from which time $\overline{A_{ot}}$ is less than the area of the thermally-eroded ice tunnel. Final draining of the tunnel occurs from about 94 s where the 4th-order polynomial shows a down turn. Further drainage (not recorded) would likely be similar to that recorded for the unperturbed hydrograph. Although the complete variation in tunnel cross-sectional area through time is complex, due to the perturbation, the general trend is commensurate with the form of a fourth-order polynomial, as illustrated more clearly for the unperturbed hydrograph (Fig. 6).

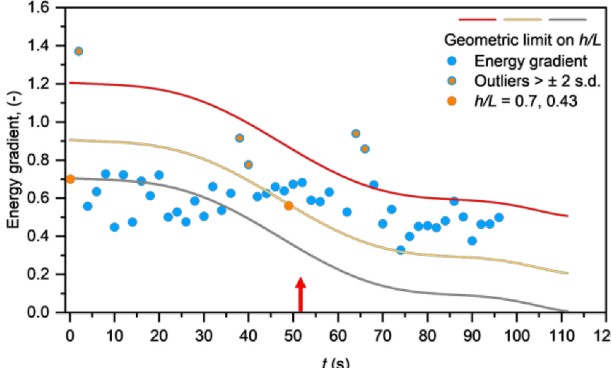


**Figure 8. Energy gradient calculated using Eq. (7), as a function of time.** The grey curve represents the geometric limit on $h/L$ if the base of the ice tunnel is constant at 0.25 m above the flume base, which is the initial height of the tunnel entrance. The yellow curve represents the geometric limit on $h/L$ if the base of the ice tunnel is constant at 0.15 m above the flume base, which is the approximate height of the base after down-cutting until the flow in the tunnel becomes aerated, with a free water

surface. The red curve represents the geometric limit on $h/L$, if the base of the ice tunnel is equal to the base of the flume – which occurred at an unknown time > 115 s. Orange dots indicate the initial $h/L = 0.7$, and when $h/L = 0.56$; i.e. just before the tunnel aerated fully (red arrow) after $t = 52$ s.

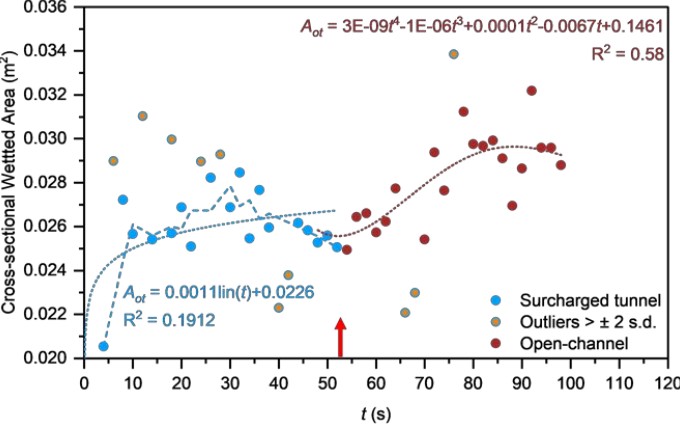

**Figure 9. Increase in the cross-sectional area ($A_{ot}$) of flow with time.** Dashed blue line indicates initial data trend that is

defined by an initial rapid logarithmic increase in tunnel area during initial surcharge (blue dotted curve). Data scatter precludes defining a simple continuous function to link the initial logarithmic function to a 4th-order polynomial latterly (brown dotted curve). Tunnel aeration increases from the (red arrow) after $t = 55$ s. Fitted functions exclude outliers.



### 3.3 Effect of water temperature variation

Comparing the hydrographs for the discharge at 25.2 °C and 12.6 °C it took longer for the discharge and flow

velocity to peak, and the maximum discharge and maximum flow rate also were smaller for the cooler water temperature. Specifically, for the 25.2 °C case, the maximum velocity was 0.25 m s⁻¹, the maximum discharge was 0.0114 m³ s⁻¹ (Fig. 10) and the time to reach the maximum values is around 50 s. In contrast, for the 12.6 °C case, the maximum velocity was 0.21 m s⁻¹, the maximum discharge was 0.0069 m³ s⁻¹ and the time to reach the maximum values is 90 to 100 s (Fig. 10). The difference in the timing of the peaks in discharge is significant

because, in nature, there would be important implications for natural flood modelling, early warning and damage mitigation procedures.

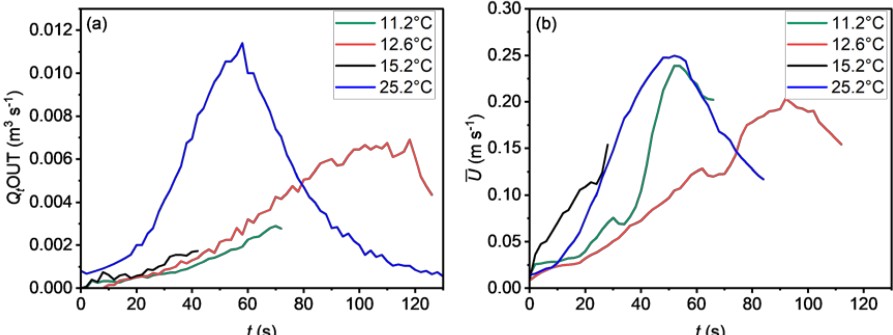

**Figure 10. Comparison of flow velocity and discharge curves at different temperatures** — the 12.6 °C discharge curve was truncated at 126 s (see main text). Both velocity curves are truncated due to the ADV becoming emergent.

### 3.4 Shape of ice tunnel

In the case of the 100 s experiments at 23.5 °C (Figs. 11 and 12a), down-cutting of the base of the tunnel was greatest towards the outlet of the tunnel. As the tunnel began to aerate, down-cutting left the upper portion of the tunnel as a quasi-ovoid and introduced a further narrower channel containing open-channel flow at the base (Fig. 11). In addition, a little water continued to run along the base of the tunnel after each flood wave had passed, due

to continued ice melt, as would occur in nature. Thus, the post-aeration discharge resulted in the formation of a vertical slot in the base of the tunnel. The foam mould reproduced both the ovoid form of the tunnel that formed under surcharged flow conditions, mediated by the vertical slot, resulting in a keyhole shape overall. Nevertheless, the overall tunnel cross-sectional area varied little within the central section with two estimates providing an average area, $\overline{A_c}$, of 122 cm².

In the case of the 126 s experiments at 12.6 °C (Fig. 12b), down-cutting of the base of the tunnel also was greatest towards the outlet of the tunnel. However, because the flood was stopped abruptly by blocking the outlet there was little post-flood down-cutting. The foam mould reproduced the quasi-ovoid form of the tunnel that formed under surcharged flow conditions. As the tunnel began to aerate, down-cutting left the upper portion of the tunnel as a quasi-ovoid and introduced a further semi-circular open-channel below leading to an overall

'pinched' ovoid or lemniscate form (Fig. 12b). The overall tunnel cross-sectional area increased slightly in the central section with distance down the tunnel, increasing from 53 cm² through 57 cm² to 72 cm²; a tunnel cross-section on average 50 % of the cross-sectional area of the 23.5 °C runs. Reducing the temperature to 11.2 °C (Fig.


12c) and a run time of 75 s resulted in a significantly smaller cross-section than was found for the roughly comparable temperature of the 126 s experiment. Similar result was obtained with a temperature of 15.2 °C and

given the short run-time (46 s) the tunnel still retained its circular appearance (Fig. 12d). It is evident that tunnel cross-sectional area is highly sensitive to water temperature and possibly the run times.

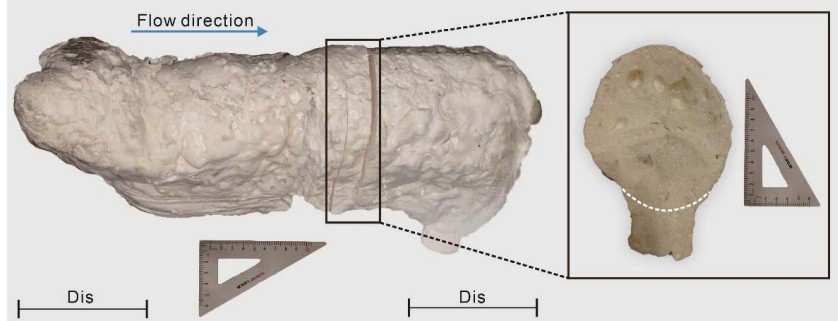

**Figure 11. The cross-sectional shape of tunnel in mid-section obtained from foam adhesive cast after 102 s of water flow during a 23.5 °C unperturbed discharge experiment.** Scales graduated in cm and mm. Dis = disturbed sections of the

tunnel which were not considered for analysis. White dotted curve indicates approximate lower limit to the surcharged tunnel before final down-cutting by post-flood drainage. The area, $A_c$ , of the ovoid tunnel cross-section as illustrated is c. 125 cm$^2$ and the basal slot below the white line adds a further few square centimetres.

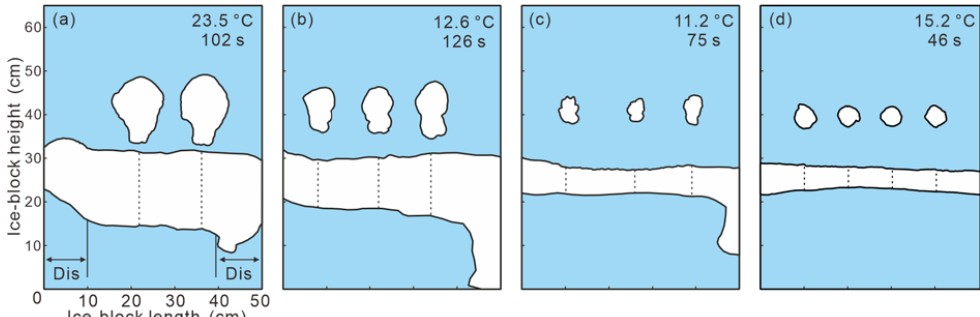

**Figure 12. Tunnel shape after floods derived using the casting technique.** Blue colour represents ice and the white areas

represent the longitudinal tunnel after the floods, with selected cross-sections shown above (in white). Black dotted lines indicate the cross section locations. Only centrally placed cross-sections were sampled, as the shape of the entrance and the exit portions of the tunnel are influenced by both the local inflow and outflow characteristics. Dis = disturbed sections of the tunnel which were not considered for analysis.

**Table 1. Observed ($A_c$) and modelled ($A'$) cross-sectional areas of the ice tunnels**

| Water temperature T (°C) | Time $t$ (s) | Sectional area, $A_c$ (cm$^2$) | | | | Average section area $\overline{A_c}$ (cm$^2$) | Simulated: $n = 0.016$ $A'$ (cm$^2$) | Simulated: $n$ variable $A'$ (cm$^2$) |
|---|---|---|---|---|---|---|---|---|
| | | $A_c$ 1 | $A_c$ 2 | $A_c$ 3 | $A_c$ 4 | | | |
| 15.2 | 46 | 19.76 | 16.86 | 15.93 | 15.02 | 16.89 | 12.60 | 12.19 |
| 11.2 | 75 | 20.51 | 15.1 | 20.99 | | 19.87 | 13.65 | 14.97 |
| 12.6 | 126 | 38.72 | 40.4 | 40.04 | | 42.72 | 45.64 | 44.12 |
| 23.5 | 102 | 96.71 | 102.36 | | | 99.53 | | |


375       The relationship of the data in Table 1 were explored using a multiple regression analysis of the form: $A_c = a + b_1 T + b_2 t$, where n = 12. Despite the limited number of independent data pairs, the analysis indicated that the water temperature explained 96 % of the variance in the value of $A_c$ with only a minor improvement (2 %) in the prediction when the duration of the experiment is included.

### 3.5 Modelling the observed discharge hydrograph

The results of implementing the simplified Nye (1976) model (Carrivick *et al.*, 2017) of the observed hydrograph behaviour are presented within Fig. 13. The parameter values used to implement the model are given in Table 2. The initial model fits were achieved using a constant value of Manning's $n = 0.0160$ m$^{-1/3}$ s (Fig. 13a). This value of $n$ was obtained iteratively in the model fitting procedure after anticipating that initial $n$ would be somewhat greater than the minimum observed form roughness: 0.008 m$^{-1/3}$ s. As has been noted prior (Nye, 1976; Carrivick

*et al.*, 2017) the fixed-$n$ model reproduces well the initial rise of the hydrograph but the similitude becomes less good closer to the peak discharge; the overall model fit being described by the mean absolute error (MAE) which should be minimized. The timing of the peak discharge is well simulated but the model over-predicts the peak discharge (Fig. 13a) indicating that the roughness of the tunnel is likely to be increasing through time, as was observed in the experiments. It is evident from the simulation of the reservoir drawdown curve that the deviation

of the model from the observed behaviour is progressive; over-predicting the rate of drawdown (Fig. 13b).

**Table 2. Variable and constant parameter values used to implement model fitting**

| Variable values | | | | |
|---|---|---|---|---|
| Water temperature, T (°C) | 12.6 | 15.2 | 23.5 | 24.0 |
| Thermal conductivity of water, $K_w$ (W m$^{-1}$ K$^{-1}$) | 0.583 | 0.589 | 0.603 | 0.604 |
| Dynamic viscosity of water, $v_w$ ($10^{-3}$ kg m$^{-1}$ s$^{-1}$) | 1.223 | 1.139 | 0.922 | 0.914 |
| Manning roughness[1], $n$ (m$^{-1/3}$ s) | 0.0160 (0.0148-0.0212) | 0.0160 (0.0158-0.0234) | 0.0160 (0.0155-0.0302) | 0.0160 (0.0144-0.0295) |
| Water depth, $h$ (m) | 0.5941 | 0.5953 | 0.5926 | 0.6024 |
| Height of conduit, $h_c$ (m) | 0.25 | 0.25 | 0.25 | 0.25 |
| Length of conduit, $l_c$ (m) | 0.50 | 0.50 | 0.50 | 0.50 |
| Initial cross-sectional area of the tunnel, $A_0$ ($10^{-5}$ m$^2$) | 7.854 | 7.854 | 7.854 | 7.854 |

| Constant values | |
|---|---|
| Ice density, $\rho_i$ (kg m$^{-3}$) | 917 |
| Water density, $\rho_w$ (kg m$^{-3}$) | 1000 |
| Specific heat capacity of water, $C_w$ (J kg$^{-1}$ K$^{-1}$) | $4.22 \times 10^3$ |
| Gravity, $g$ (m s$^{-2}$) | 9.8 |
| Latent heat of melting of water, $L_w$ (J kg$^{-1}$) | $333.5 \times 10^3$ |

[1]Manning's roughness value used for modelling with constant value (values used for modelling with variable $n$).

      To adjust Manning's $n$ through time it was reasoned that the change in the cross-sectional area of the tunnel through time would be proportional to any change in form and shape roughness. An equation of the form: $n =$

$a + b(A'-A_0)/A_0$ was found to be appropriate; where $A_o$ is the initial tunnel area at $t = 0$ and $A'$ is the modelled cross-sectional area. The coefficients $a$ and $b$ were adjusted iteratively to achieve the best fitting relationship, such that as time ($t$) cumulated during the hydrograph, $n$ was recalculated to increase incrementally within the model as $A'$ increased. As a change in model $n$ also will affect the anticipated future value of $A'$ at the next time step, the two




values are in a relationship of dynamic adjustment. Further consideration would be useful with respect to whether

the dynamic adjustment of the tunnel cross-sectional area is the best means to predict increments in $n$. For example, the time increment can be used to iteratively update the values of $n$ in the model. This latter approach (not illustrated) gave MAE values only slightly worse than those presented. However, time is not an attribute of the tunnel roughness and so the relationship of $n$ with $A'$ is to be preferred. This procedure resulted in calculated values of $A'$ being consistent with the observed average values of $A_c$ (Fig. 14).

Thus the model fits were adjusted using an initial value of Manning's $n = 0.0144$ m$^{-1/3}$ s rising to 0.0295 m$^{-1/3}$ s (Fig. 13c) as the hydrograph progressed. The similitude is much improved close to the peak discharge; the mean absolute error (MAE) being reduced in contrast to Fig. 13a. It is evident from the simulation of the reservoir drawdown curve that the deviation of the model from the observed behaviour remains progressive but is much reduced (Fig. 13d). As might be expected for a model designed to simulate surcharged pipeflow, the model

discharge deviates from the observed discharge once the conduit is no longer surcharged.

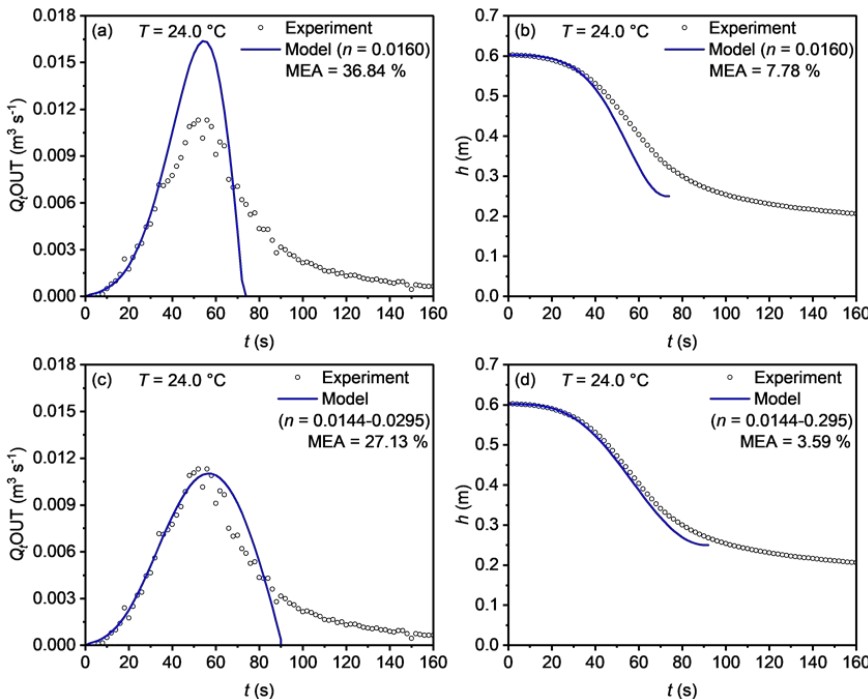

**Figure 13. Examples of model fit to hydrograph and drawdown curve for a water temperature of 24.0 °C.** (a) Model (blue) hydrograph for constant value of Manning's $n$; (b) Model drawdown (blue) curve for constant value of Manning's $n$; (c) Model (blue) hydrograph for time-varying value of Manning's $n$; (d) Model drawdown (blue) curve for time-varying value of

Manning's $n$.





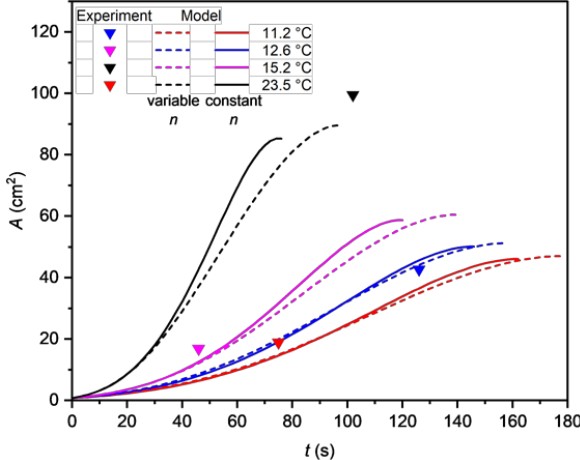

**Figure 14.** Time trends of the calculated values of the cross-sectional area ($A'$) in comparison with the observed average values of $A_c$.

## 4 Discussion

The purpose of the experiments was to gain an improved understanding of how simple linear ice tunnels grow when subject to sudden time-varying discharge. Of particular interest are the rate of change and peak discharge as water temperature was varied. As expected (Spring & Hutter, 1981), water temperature had a major influence on both the rate of change of discharge on the rising limb of the hydrograph and on the peak discharge, with discharge increasing near monotonically, which is typical of such floods (Nye, 1976). The lack of model fit on

the falling limb has been noted in many earlier attempts to model GLOFs (reviewed by Carrivick *et al*, 2017). Fowler (2009) commented that this may be due to tunnel collapse which would both perturb the hydrograph and lead to an altered tunnel shape. Localized collapse is unlikely to lead to a systematic and temporally-prolonged change in the falling limb of the hydrograph. Rather, the application of a model for surcharged conduit flow become less appropriate as the tunnel begins to aerate. Herein, for brevity, we do not consider further the controls

on the falling hydrograph, other than to note that abrupt steep recession curves have been related to the presence of an over-pressurized broad lateral fracture (*i.e.* non-circular section) (Fowler, 2009) which is evidently not the case with our prolonged recession curves.

In addition, with respect to modelling tunnel-confined GLOFs, the evolution of the energy slope is pertinent to potential GLOF modelling frameworks. Although the energy slope data are inevitably noisy, Equation (7)

successfully predicted the energy slope of the tunnel discharge within the geometric constraints imposed by the experimental design. However the cross-sectional areas of the 25°C tunnels, located mid-dam, as derived using the expanding foam, are somewhat less than the cross-sectional area of the entrance derived from Eq. (5), as used in Eq. (7). Thus, it is evident that as the tunnel entrance increases in diameter the velocity ($\bar{U}$) measured at $x_o = 0$ + ε is an under-estimate of the average velocity throughout the tunnel where the tunnel is smaller in area, until the

tunnel expands at the exit. The eroded tunnel shape is conditioned by the initial choice of a circular tunnel, that then developed an ovoid form and latterly a 'keyhole' shape. The choice of a circular channel was related to the often assumed tunnel shape (Björnsson, 1992; Fowler, 2009) although Hooke *et al*., (1990) argued a segment of



a circle might be a more appropriate (with the arc as the base), especially where the base of the tunnel is rock debris. In our experiments, the keyhole shape developed because the tunnel base remained within pure ice during

most of the time. In this respect as tunnel area increases due to frictional heating, which will be applied fairly evenly around the perimeter, an ovoid form for surcharged tunnels is not inappropriate.

The average velocity and the distribution of the flow within the main tunnel remains unknown. Attempts to derive an average velocity within the tunnel by varying the velocity within Eq. (7) to reduce the expected cross-sectional area of the tunnel ($A_{ot}$) to equate to that derived from the foam casts ($A_c$) was unsuccessful; excessively

high energy gradients were obtained with unreasonably low roughness coefficients. This outcome is not unexpected, as Barr (1981) specifically noted that obtaining accurate tunnel areas from velocity data alone was difficult. Equation (7) was developed using data for metal pipes of fixed diameter and fixed roughness, so limitations can be expected with applications to the dynamic enlargement of an ice conduit. For example, tortuous pipes require consideration of the shape drag as well as skin and form drag (Morvan *et al*., 2002). Although this

additional consideration might not be considered significant in the relatively straight tunnels considered herein, the shape drag might be highly significant in tortuous natural sub-glacial tunnels draining ice-dammed lakes (Nye, 1976).

The values of $k_s$ require comment. There appears to be no literature on the direct calculation of Nikuradse's equivalent roughness for water flow through ice tunnels. As the tunnel initially is surcharged, strictly there is no

justification to use the open-channel Manning's *n* as the roughness parameter, although this is a required input to the Nye model. Nevertheless, a consideration of the conversion of $k_s$ to Manning's *n* using equations relating grain roughness for sand-lined pipeflow (Butler *et al*., 1996; Marriot & Jayaratne, 2010) returns initial 'skin' Manning's *n* values of around 0.002 m$^{-1/3}$ s rising to 0.008 m$^{-1/3}$ s as form roughness developed. Accepting a general analogy of ice asperities to sand grain roughness in pipes is valid, and in the absence of direct measures of tunnel skin ice

roughness, the larger values of Manning's *n* are comparable to those reported in the literature for glass (0.009 m$^{-1/3}$ s) and lucite (0.008 m$^{-1/3}$ s), which thus may be appropriate analogues for relatively smooth ice.

In an application of the simplified Nye model to a field example of straight ice tunnel development, Carrivick *et al*. (2017) explored the range of suitable values of *n*, and found that the minimum in their application was around $n = 0.005$ m$^{-1/3}$ s. This value is central to our direct observations of the range of value of *n* for smooth ice.

Nevertheless, the drainage model of Carrivick *et al*. (2017) produced good agreement between their predicted discharges and their observed discharges when Manning's *n* was $< 0.2$ m$^{-1/3}$ s, with an optimal range of 0.03 to 0.045 m$^{-1/3}$ s. Thus the simulations of Carrivick *et al*. (2017) indicate that considerable form and shape drag components develop during natural discharge events, additional to that drag component which can be calculated from the Nikuradse equivalent (skin) roughness component. Our experimental data confirm the necessity to

increase the model tunnel roughness as the hydrograph progresses, in that the best-fit Manning's *n* values in our experiments ranged between 0.0144 and 0.0302 m$^{-1/3}$ s, increasing as the hydrographs progressed and the tunnels enlarged. These latter Manning's *n* values are consistent with the values of *n* used in some of the more recent models of natural GLOFs (reviewed by Carrivick *et al*., 2017) and 0.03 m$^{-1/3}$ s was recommended by Clarke (2003) as a suitable value. However, models of several natural GLOFS, using a fixed value of *n*, have required

significantly larger values (Hooke *et al*., 1990; Björnsson, 1992; Fowler, 2009) which may reflect extreme tunnel complexity or the presence of debris on the bed (Fowler, 2009). However, it is well-known in modelling open-channel hydraulics that large values of *n* often compensate for model inadequacies rather than reflect channel



roughness. Importantly, the similarity of the range of Mannings $n$ values found in the present experiments and those reported for natural GLOFs, indicates a degree of scale-independence, such that Reynolds-number-scaled physical experiments on ice-dam tunnel flow replicate aspects of larger scale natural outbursts.

An alternative experimental design would be required to obtain mid-tunnel velocity and roughness data to aid future numerical modelling. If there was an opportunity to measure the velocity of a natural tunnel-confined GLOF emanating from a glacier, it is more likely that this could be measured within the lake at the tunnel entrance, or at the exit at the ice front, rather than deep within the tunnel. However, dye-tracing has promise to obtain system-averaged velocities (Werder & Funk, 2009). In this respect, it would be important to relate the often larger diameter entrance and exit sections to a probable smaller section deep within the tunnel. Nevertheless, although the melt-rate will vary somewhat as the tunnel circumference varies, the growth of the cross-sectional area of the tunnel primarily was controlled by the water temperature and the tunnel section increased rapidly half-way through the rising hydrograph to accommodate the discharge as wall roughness increased and the rate of velocity increase declined. Carrivick *et al.*, (2017) also noted a rapid increase in tunnel enlargement was required to sustain discharge in their model of a natural GLOF. Indeed, the observed cross-sectional areas and the areas calculated using the Carrivick *et al.* (2020) simplified Nye-model are excellent (Table 1). Consequently, the present results provide a useful framework to constrain both the behaviour of the energy slope, the range of roughness values experienced within a tunnel and the growth behaviour of the tunnel cross-section as the GLOF hydrograph progresses.

## 5 Conclusions

Controlled laboratory experiments can shed light on the process of ice-tunnel enlargement associated with GLOFs. The behaviour of the rising hydrograph in our experiments is consistent with theory, being near monotonic. The water temperature was related positively to the rate of rise of hydrographs. For any water temperature, and a circular tunnel exhibiting primarily skin roughness only, the surcharged wetted tunnel cross-section increased as a logarithmic function of time with velocity also increasing on the rising hydrograph. As frictional melt induced form roughness, velocity declined and the surcharged tunnel cross-sectional area increased to maintain the discharge. On the falling hydrograph limb, once a free surface developed the decline in the open-channel wetted area was linear. The pipeflow equation proposed by Barr in 1981, described well the behaviour of the energy slope with an initial tunnel Nikuradse equivalent roughness of $c$. $10^{-9}$ increasing to $10^{-4}$ m as the hydrograph progressed. An ovoid tunnel section (long axis vertical) rapidly evolved from the initial circular section as the discharged increased during surcharged conditions. However, down-cutting was rapid once a free surface developed which led to a final key-hole shape to the tunnel.

## Data available

The original experiment data of this article are available for formal review through a link provided to the editorial office. If the manuscript is accepted for publication, the data will be archived on a publically available platform with DOI reference.





**Author contributions**

Paul A. Carling proposed the experimental design, and Chengbin Zou and Xuanmei Fan designed the experimental
facility and conducted experiments. Chengbin Zou performed the numerical simulations with assistance from
Zetao Feng. Daniel R. Parsons supplied technical equipment. The data were analysed jointly by Paul A. Carling
and Chengbin Zou. Paul A. Carling prepared the manuscript with contributions from all co-authors.

**Competing interests**

The authors declare that they have no conflict of interest.

**Acknowledgements**

Thanks to Tao Wei and Xin Wen for the assistance in experiments.

**Financial support**

This research is financially supported by the National Science Fund for Distinguished Young Scholars of China
(grant no. 42125702). Parsons was supported via a European Research Council award (GEOSTICK; Grant
agreement 725955).

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
