# Peer review of "Physical Experiments on the Development of an Ice Tunnel from an Upstream Water Reservoir through Simulated Glacier Dam"

_The Cryosphere, 2022_

## Referee Comment (RC2)

This study uses laboratory experiments to investigate ice-walled channel growth during glacier outburst floods. A reservoir of water sits adjacent to an ice block with an initially circular channel within the ice. The supply of water drains through the channel and frictional melting causes the channel walls to melt and for the shape and cross-sectional area to evolve in time. Discharge rates into and out of the channel are recorded along with the velocity at the entrance of the channel and water depth at the reservoir. Water temperature, duration of drainage, and additional water added to reservoir are varied between experiments to test the influence of these variables on discharge, energy gradient, and cross-sectional area. Water temperature is found to have a greater effect on cross-sectional area than the duration of the drainage. Comparing modeled and observed results suggest that hydraulic roughness increases (due to an increase in form drag) as a flood evolves. Although I think that the approach and results are really interesting and have important implications for glacier outburst flood models (and glacier hydrology models in general), some aspects of the study were unclear and the manuscript could benefit from some re-structuring. Below I list some general concerns (in no particular order) and then some line-by-line comments.

**General concerns**

I see the primary contribution from this project being the results that suggest that the hydraulic roughness increases as a flood evolves due to the creation of form drag. I recommend changing the title to "Laboratory experiments indicate increasing hydraulic roughness during glacier outburst floods due to the creation of form drag" or something that alludes to this point. This would make the key message of the paper immediately clear. I also recommend making this point more clear by discussing the previous work on hydraulic roughness in the introduction.

There are some curious aspects of the outburst floods that are not fully explained – the rising limb appears more rapid than I would anticipate for this type of flood, and the falling limb appears much longer. What was the initial conduit size relative to the ice block, and how might this compare to real systems? Was the initial conduit size relatively large? Furthermore, in the experiments the conduit size didn't change shape at first, which had me wondering whether the ice was well below freezing. Perhaps heat was initially going into warming the ice instead of melting, which could explain why the conduit didn't grow very quickly at first. Slower conduit growth may also have prevented the conduit from "over shooting" and causing a very rapid drop in discharge at the end of the flood. This is speculative, but without further details one is left wondering why the experiments different from theory.

The authors state in lines 133–135, 'Water temperature of ice-dammed lakes often is assumed to be close to zero although temperature of Icelandic GLOFs have been measured, for example, as 0.05 °C (Rist, 1955), 0 to 1°C (Einarsson et al., 2017) and up to 10 °C (Carrivick et al., 2020).' The temperatures used in the study ($11.2 - 25.2°$ C are up 15° C warmer than that observed in natural GLOFs. Other than the sections discussing the effects of temperature and the shape of the ice tunnel, 3.3 and 3.4 respectively, the results focus on temperatures $24 - 25.2°$ C. The lowest and most realistic water temperature considered in this study is 11.2° C, but this temperature was not used in most of the data analysis. Because these higher temperatures are not realistic, there needs to be a discussion on how this may change the findings and results of the study. For example, the authors state in lines 483-484 that the range of Mannings $n$ values in this study are similar to those of natural GLOFs which indicates a degree of scale-independence. This statement does not

acknowledge the effect that temperature may have on the results.

The cross-sectional areas of the channels with four different water temperatures are measured at all significantly different drainage times $(46, 75, 102,$ and $126$ s). The choice to terminate flow for different times and temperatures makes it difficult to see the influence of time vs. temperature on cross-sectional area in Table 1. In the line 376, the authors state that $n = 12$, but there are only four samples shown in Table 1. If there is more data not listed in the table, mention this in the caption or in the text.

Hydraulic gradient $I$, energy slope, and energy gradient are all used interchangeably which is slightly confusing. The terminology may be unfamiliar to glaciologists (especially energy slope and energy gradient) and should be clarified.

When, and why, was an additional pulse of water added to the system? This should be included in the Methods section.

Define and discuss Nikuradse equivalent roughness and 'skin roughness' somewhere in the Introduction (Section 1) or Methods (Section 2) and how this relates to Manning's roughness. I was unfamiliar with the term and suspect that other glaciologists may also be unfamiliar with it.

Why was a linear reservoir model used (Eq. 1-4) instead of calculating the discharge out of the reservoir by multiplying $dh/dt$ by the area of the reservoir? Is this because of the melt occurring at the ice dam? If so, mention this.

A more thorough description of the model should be included in Section 3.5. Adding Eqs. (1) and (2) from Carrivick et al. (2017) would be helpful. It would also be nice to see more details about how relevant dimensionless numbers are calculated for these experiments (even if it is only included in an appendix). For example, what is the conduit aspect ratio and how does it compare to real conduits?

It is sometimes hard to follow whether the the authors are discussing the rising or the falling limb of the outburst flood. This needs to be more clear throughout the paper.

**Line by line comments**

9: Is this study relevant to proglacial lakes? Ice-dammed or ice-marginal glacier lakes in general would be better here. This work is also relevant to glacier hydrology in general and this could be emphasized in the introduction.

24: Change to say, 'For any given temperature or skin roughness condition'. This includes the different combinations of temperature and roughness condition.

25-26: This line sounds as though the induced form roughness is the cause of the increase cross-sectional area and decreasing velocity. Since the velocity declines only after the cross-sectional area increases, switch the order of these two. 'As frictional melt induces form roughness, the surcharged tunnel cross-sectional area increases to accommodate the discharge and velocity declines.'

33: Again, 'proglacial lakes' should be changed here.

56: Flood water can be at the freezing point. Add to say 'temperature of flood water is often above freezing point'.

62-63: Reword. Perhaps instead of saying 'closure or an increase in tunnel capacity' say 'changes in tunnel capacity'. Ice deformation is mentioned in line 60 and is sometimes referred to as closure, which makes this slightly confusing.

69-71: Combine these two sentences and reword. In line 69, the authors state 'The purpose of the experiment is several fold.' The following two sentences are justifying the choice of a straight conduit and don't give much insight to the purpose of the experiment.

92: I am not familiar with the inequality notation used here.

97: What was the ice temperature when the experiments were run? Add the ice temperature here. If it is below $0°$ C, then discuss how this would change the results in Discussion.

98: In reference to calculating the Reynolds number, add 'Later we discuss how we arrived at these estimates.'

121: Paragraph begins with limitations of the experiment. Move first sentence to later in paragraph.

131-132: The author states 'This latter observation' but no previous observation was listed (only a method).

136: 'At two different times of the year...' Times is plural in this context.

146: 'Although colder controlled temperatures...'

150: $k$ is used for recession constant and $k_s$ is used for skin roughness. It may be more clear (and not too hard to change) if the recession constant is denoted as something else.

148-156: Equation (1) is reference in line 111, but the equation does not appear until line 153. The presentation and written description of Equations (1) – (4) are confusing, especially if the reader glances at Eq. (1) when it is first mentioned. It may be easier to understand if the description is written in-line with equations.

168: Missing 'of' between 'of the base' and 'the tunnel inlet'.

177: 'so $I$ is calculated instead' is confusing after the description of how to calculate $I$ using the difference between heights and the limitations of the model setup. Perhaps instead say 'so $I$ is calculated instead using an approximation of the Colebrook-White equation described below.'

185: $k_s$ and $k_s''$ are defined but not $k_s'$

197-198: State the range of the Reynolds values in this experiment in line here.

200: Cite Table 2 where values for kinematic viscosity of water for different temperatures are listed.

203: This sounds like the flow was stopped multiple times for each experiment. This is not the case, correct?

217: Three things are listed despite stating 'both' which implies only two.

Section 3.1 and 3.2: Why was $25.2°$ C chosen over the $21.5°$ C experiment if the results were similar and $21.5°$ C is more realistic?

229: The value of $k$ is mentioned in for 58–102 seconds but no other part of the hydrograph. Either delete $k$ value here or or add to earlier times (unless there is some reason for including it).

Fig 4: The rising limb of the hydrograph appears faster than expected. Is this because the initial conduit cross-sectional area is relatively large? A discussion of scaling would be helpful.

242: Here $I = h/L$, but in line 175 the authors stated that the hydraulic gradient $I$ is the ratio of the water depth and the height of the water surface at tunnel outlet. Why is this?

244: It is not clear what $H$ represents.

251: Do you mean Fig. 5 (instead of Fig. 4)?

Fig. 10: Why is there is a time difference between the discharge and velocity curves on plots for the 15.2° C and 25.2° C? Did you record more discharge data for one experiment and more velocity data for the other?

376: It is not 'despite' the small sample size that results were obtained. How about 'Acknowledging the limited number...'?

Table 2: I would rename 'length of conduit' and 'height of conduit' to say 'initial' length and height of conduit. Then move initial cross-sectional area, initial length of conduit, and initial height of conduit under the 'Constant Values' heading. Why was the initial water depth different between experiments?

Fig 13: Denote the time in experiment when the conduit is no longer surcharged in the plots or add the time when this happens to the figure caption (or state the time in line 410 where it is referenced). The simplified Nye model does not handle open channel flow. Do the modeled curves in Fig. 13 continue past this time?

Fig 14: Are the triangles representing the peak discharge from the observed experiments? This part of legend is difficult to understand and could be described in the caption.

421: Authors state 'how simple linear channels grow when subject to sudden time-varying discharge', but the time-varying discharge is also a result of channel growth. This should be reworded to say, 'when surcharged by adjacent reservoir' or something similar.

430: Is the latter part of this sentence stating 'other than to note...with our prolonged recession curves' necessary?

441: 'The choice of a circular channel ...' This sentence seems out of place. Consider moving it to the methods section. A sentence should be added justifying why this tunnel shape was chosen over a semi-circular channel at the bed. Are englacial conduits more generalizable since basal roughness does not need to be considered? Were they easier to fabricate in the experiments?

461: Delete 'strictly'.

503: 'near monotonic'? This seems redundant since the rising hydrographs typically means the phase of the flood where discharge is monotonically increasing.

506: Repeated sentence from abstract. See comments in lines 25-26.

513: There could be a much more informative concluding sentence.